

# Climatology in Asian dust activation and transport based on MISR satellite observations and trajectory analysis

Yan Yu[1], Olga V. Kalashnikova[2], Michael J. Garay[2], and Michael Notaro[3]

[1]Department of Geography, University of California, Los Angeles, CA 90095, USA

[2]Jet Propulsion Laboratory, California Institute of Technology, Pasadena, CA 91109, USA

[3]Nelson Institute Center for Climatic Research, University of Wisconsin-Madison, Madison, WI 53706, USA

*Correspondence to*: Yan Yu (yuyan06@gmail.com)

**Abstract.** Asian dust, primarily emitted from the Taklamakan and Gobi Deserts, has been reported to reach remote destinations, such as North America. However, the relative contribution of the Taklamakan and Gobi Deserts to dust

loadings through long-range transport remains unaddressed in any observational study. Here, the climatology of Asian dust activation and transport is investigated using stereo observations of dust sources from the Multiangle Imaging SpectroRadiometer (MISR) instrument combined with observation-initiated trajectory modeling. MISR-derived dust injection height and dust plume motion vectors confirm the peak of dust activation and transport potential in spring over the Gobi Desert and in both spring and summer over the Taklamakan Desert. The long-range transport patterns of Asian dust,

including the influence on North America through trans-Pacific transport, are assessed using extensive forward trajectories initiated by MISR dust plume observations. The trajectory analysis reveals latitude-dependent influence of dust from the Taklamakan and Gobi deserts, with Taklamakan dust dominantly affecting to the south of 50˚N and Gobi dust primarily affecting to the north of 50˚N in North America. The Asian dust activation and transport exhibit substantial seasonal and interannual variability, motivating future studies on the potential drivers.

**1 Introduction**

Long-range transport of Asian dust has been documented in observational records and modeling results. It has been estimated that Asian dust sources account for 3-11% of global dust emissions (Tanaka and Chiba, 2006). Originating primarily from the Taklamakan and Gobi Deserts (Sun et al., 2001), elevated Asian dust is carried eastward by the prevailing mid-latitude westerlies and crosses over China, Korea, and Japan to the North Pacific (Huang et al., 2008; Uno et al., 2011;

Yumimoto et al., 2010). Asian dust occasionally reaches North America (Eguchi et al., 2009) and the Arctic (Huang et al., 2015), and can even be transported for more than one full track around the globe (Uno et al., 2009). Past observational studies have extensively examined individual cases of long-range, especially the trans-Pacific, transport of Asian dust in terms of the transport pathways and vertical structure of aerosols (e.g. Arimoto et al., 2006; Eguchi et al., 2009; Liu et al., 2013; Uno et al., 2011; Yumimoto et al., 2010). Several field campaigns have been conducted to examine the composition,

properties and radiative effects of Asian dust along its path of transport, such as the Asian Aerosol Characterization Experiment (ACE-Asia) (Huebert, 2003), the Intercontinental Chemical Transport Experiment (INTEX-B) (McKendry et al.,



2008), Aeolian Dust Experiment on Climate Impact (ADEC) (Mikami et al., 2006), the National Aeronautics and Space Administration (NASA) Transport and Chemical Evolution over the Pacific (TRACE-P) (Jacob et al., 2003), and the Pacific Dust Experiment (Stith et al., 2009). However, due to the limited temporal coverage, these observational case studies or field campaigns did not address fundamental long-term statistics, such as the frequency of trans-Pacific transport of Asian dust.

5  The widely spread Asian dust exerts diverse influences on the regional and global environment. East Asian dust often mixes with pollutants, such as sulfate and nitrate, during the transport over the heavily polluted regions in China (Huang et al., 2010; Liu et al., 2014; Wang et al., 2016), leading to degraded air quality downwind in China (Yang et al., 2017), South Korea (Ha et al., 2017), Japan (Tobo et al., 2010), and the United States (Wu et al., 2015; Zhao et al., 2008). In particular, based on dust mass fluxes derived from satellite Aerosol Optical Depth (AOD) observations and zonal wind

10 assimilations, Yu et al. (2012) concluded that the trans-Pacific transported Asian dust overwhelms the locally emitted dust in North America. Asian dust deposits into the marginal seas of China and North Pacific, carrying with it bioavailable iron that enhances phytoplankton growth (Tan et al., 2017; Wang et al., 2012; Yuan and Zhang, 2006). Asian dust has been widely reported to influence regional and global climate directly by altering the radiative budget, and indirectly by modifying cloud microphysics (Chen et al., 2017; Ge et al., 2008; Huang et al., 2010; Huang et al., 2014; Huang et al., 2006a,b; Li et al.,

15 2016; Wang et al., 2010). In particular, observational evidence indicates that Asian dust aerosols serve as ice nuclei and influence orographic precipitation processes over the western United States (Creamean et al., 2013).

  Despite the important role of Asian dust in the global environment, the relative contribution of Asian dust sources, namely the Taklamakan and Gobi Deserts (Fig.1), to dust activity at remote destinations, such as North America, remains unaddressed by observational studies (Chen et al., 2017). The Taklamakan Desert is located in the Tarim Basin of

20 northwestern China and bounded by the Kunlun Mountains, the Pamir Plateau, and Tian Shan Mountain to the south, west, and north, respectively. The Gobi Desert is located to the east of the Taklamakan Desert, covering portions of northern China and southern Mongolia. Based on a size-dependent soil dust emission and transport model, namely the Northern Aerosol Regional Climate Model (NARCM), Zhang et al. (2003) concluded that the Gobi Desert emitted about twice the amount of dust as the Taklamakan Desert during spring of 1960-2002. Based on the Weather Research and Forecasting Model with

25 chemistry (WRF-Chem) simulation of the massive East Asian dust storm in March 2010, Chen et al. (2017) revealed comparable dust emission fluxes from the Gobi and Taklamakan Deserts but suggested the higher potential for long-range dust transport from the Gobi Desert where the surface wind is dominantly eastward and the ground has higher altitude than the Taklamakan Desert. Given the substantial uncertainty in simulating dust emission and transport, due to difficulties in the parameterization of wind speed, soil water content, and vegetation cover (Uno et al., 2006), the modeling results regarding

30 the relative contributions of the Taklamakan and Gobi Deserts toward dust emission and transport need to be verified using observational data. Moreover, in light of the distinguished size distributions, chemical compositions, and optical properties of dust aerosols from the two deserts, knowledge about the dust activation and transport from the two deserts will facilitate better understanding of the remote environmental influence of Asian dust.



In the current study, dust source activities across the Taklamakan and Gobi Deserts are examined using stereo retrievals of dust plumes observed by the Multi-angle Imaging SpectroRadioeter (MISR) (Diner et al., 1998) instrument on the NASA polar-orbiting Terra satellite. The plume heights and vector winds are retrieved by the MISR Interactive eXplorer (MINX) tool (Nelson et al., 2013) in case-by-case plume analysis. The multi-angle capability of MISR facilitates the

stereoscopic retrieval of heights and motion vectors for clouds and aerosol plumes (Moroney et al., 2002). By incorporating additional information on the direction of apparent plume motion from a trained user, the MINX visualization and analysis software enables precise retrievals of aerosol plume heights and instantaneous winds at a horizontal resolution of 1.1 km. In prior studies, MINX plume height and motion retrievals provided unique and valuable information on aerosol injection heights from volcanic eruptions (Flower and Kahn, 2018), fires (Val Martin et al., 2010), and dust source activations

(Kalashnikova et al., 2011). The dust plume measurements from MINX enable the current examination of the climatology in dust injection heights, which is a key parameter determining the potential for long-range transport yet has never been examined in observational data, across the Taklamakan and Gobi Deserts. Furthermore, precise observations of dust injection heights allow more accurate trajectory modeling for the investigation of dust transport.

Through an analysis of plume height and motion observations from MISR and application of trajectory analysis, the

present study investigates the climatology in dust source activity and dust transport from the Taklamakan and Gobi Deserts. In particular, the following questions are addressed. Which dust source, between the Taklamakan and Gobi Deserts, is a greater contributor towards the long-range transport of dust, such as the trans-Pacific transport to North America? How often do the trans-Pacific dust transport events occur? Is there any seasonality and/or interannual variability in the dust activation and transport?

## 20   2. Data and method

### 2.1 MINX retrieval of dust plume height and motion

Dust plumes from the Taklamakan and Gobi Deserts were processed with the MINX tool. Using multiple MISR imagery, the trained user of the MINX tool derives aerosol plume top height and wind at a spatial resolution of 1.1 x 1.1 km, with estimated uncertainties in height and wind speed of 200 m and 1-2 m s$^{-1}$, respectively (Nelson et al., 2008). The MINX-based

plume height and motion measurements are obtained geometrically, independent of background aerosols and thin cirrus, atmospheric thermal structure, cloud emissivity, or instrument radiometric calibration (Kalashnikova et al., 2011). The MISR instrument, with its 380 km swath, views the study region of East Asia every 6-7 days. In the present study, the MINX dust plume data includes 2,251 dust events with 310,290 dust plume data points across the central Gobi Desert (36˚N-45˚N, 90˚E-120˚E) during 2001-2003, and 8,945 events with 10,867,131 dust plume data points across the Taklamakan Desert (36˚N-

42˚N, 77˚E-91˚E) during 2001-2011. Given the extensive labor work involved in the MINX retrieval, we currently only have three years of data over the Gobi Desert and 11 years of data over the Taklamakan Desert. However, the multi-year MINX dust plume data provide unique and sufficient information on the long-term statistics of dust plume characteristics.



According to the spatial distribution of MINX dust plume samples (Fig.1), substantial dust activation occurs across the entire Taklamakan Desert, with the highest frequency over the Hexi Corridor, where the surface wind is intensified due to the tunnel effect by surrounding mountains. In contrast, across the Gobi Desert, dust activation is limited to several hotspots, such as to the lee of the Yin, Qilian, and Altai Mountains, and over the Ordos Desert.

## 2.2 HYSPLIT forward trajectory analysis

In order to identify the transport pathways of dust emitted from the Taklamakan and Gobi Deserts, here we apply the Hybrid Single-Particle Lagrangian Integrated Trajectory (HYSPLIT) (Stein et al., 2015) model from the National Ocean and Atmosphere Administration (NOAA) Air Resource Laboratory. The HYSPLIT forward trajectory analysis has been widely used for tracing the downwind evolution of Saharan and Asian dust (Guo et al., 2017; Salvador et al., 2014; Su & Toon, 2011; Wang et al., 2016). Given the spatially widespread influence of Asian dust, long trajectories that last for weeks have been analyzed in previous studies (Guo et al., 2017; Huang et al., 2015; Uno et al., 2009; Wang et al., 2013). In the present study, 14-day forward trajectories from the Taklamakan Desert during 2001-2011 and from the Gobi Desert during 2001-2003 are computed based on six-hourly, three-dimensional wind fields on a 2.5˚ x 2.5˚ grid from the National Centers for Environmental Prediction (NCEP)-National Center for Atmospheric Research (NCAR) Reanalysis. In recognition of the coarse spatial resolution of the NCEP-NCAR Reanalysis, forward trajectories from the Taklamakan Desert are also computed based on wind fields from the NCEP Global Data Assimilation System on a 1˚x1˚ grid (GDAS1) during 2006-2011 to confirm the conclusions from the trajectory analysis driven by the NCEP-NCAR reanalysis.

The initial date, time (around 5 am UTC), latitude, longitude, and height of trajectory are obtained from MINX dust plume observations. Corresponding to the available MINX dust plume observations, a total of 310,290 forward trajectories are initiated from the 2,251 dust events across the central Gobi Desert (36˚N- 45˚N, 90˚E-120˚E) during 2001-2003, and 10,867,131 forward trajectories are initiated from the 8,945 dust events across the Taklamakan Desert (36˚N-42˚N, 77˚E-91˚E) during 2001-2011. Given the spatial resolution of 1.1 km of the MINX dust plume data, the trajectories initiated from the nearby points in the same dust plume constitute a natural ensemble (Fig.S1), thereby minimizing the trajectory calculation error. By initiating the trajectory model with the observed, precise injection height, the modeled transport pathways are improved.

In order to explore the sensitivity of atmospheric suspension time to initial injection height over the two deserts, experimental trajectories from the dust emission hotspots in the Taklamakan (40˚N, 89˚E, elevation = 805 m) and Gobi Deserts (43.5˚N, 130˚E, elevation = 954 m) were initialized with injection height varying from 100 m to 4000 m above ground in the previously reported dust active months of March-May during 2001-2003 (Fig.2). According to this experiment, the atmospheric suspension time is sensitive to the initial injection height in the trajectory model for dust particles emitted from both the Taklamakan and Gobi Deserts. Dust particles that are injected higher into the atmosphere generally have higher potential for longer atmospheric suspension time and thereby permit transport over a longer distance. In particular, particle suspension time is most sensitive to initial injection height when the injection height is below 2000 m above mean



sea level (ASL) over the Taklamakan Desert and 2500 m ASL over the Gobi Desert, potentially attributed to the differentiated vertical profile of atmospheric stability over the two regions indicated by the climatology in vertical motion (Fig.S2). Incremental increases of 100 m in initial injection height over both deserts lead to increases in the median atmospheric suspension time by more than one week at the most, suggesting the critical role of precise injection height in

accurate modeling of dust transport and thereby determining the long-range transport. The sensitivity of atmospheric suspension time to injection height appears to be independent of the meteorological driver of the trajectories, since quantitatively similar sensitivity is evident in the analysis based on additional trajectories driven by GDAS1 and NCEP-NCAR during 2006-2008 (Fig.S3).

### 2.3 MISR wind and MERRA2 reanalysis

In order to identify the dust transport directions over the Taklamakan and Gobi Deserts, the wind climatology is examined at different vertical levels at a spatial resolution of 17.6 km from version F02_0002 of the MISR Level 3 Cloud Motion Vector Product (CMVP) for the time period of 2000-2017. In this product, imagery from multiple MISR cameras is used to simultaneously retrieve motion, namely wind speed and direction, and top height of automatically tracked features, such as optically thick aerosol plumes and clouds, thereby providing a proxy measure of wind (Mueller et al., 2012). In data-sparse

regions like the Taklamakan and Gobi Deserts, CMVP provides valuable information on the observed wind and complements any reanalysis or model simulation. Compared with other atmospheric motion vectors (AMVs) products based on radiometric heights, such as those retrieved by Geostationary Operational Environmental Satellite (GOES) and Moderate Resolution Imaging Spectroradiometer (MODIS) AMV algorithms, the geometric heights assigned to MISR CMVs are not sensitive to radiometric calibrations, thereby facilitating higher accuracy (Mueller et al., 2017). MISR CMVP has been

successfully applied to identify dust sources associated with the high frequency of rapidly propagating dust plumes near the surface across North Africa and the Middle East (Yu et al., 2018). In addition to MISR wind, vertical motion from the 0.5° x 0.625° Modern-Era Retrospective analysis for Research and Applications, Version 2 (MERRA-2) (Gelaro et al., 2017) was also analyzed in an effort to explain the difference in the injection height and potential for long-range transport of dust from the Taklamakan and Gobi Deserts, and confirm the findings from trajectories driven by NCEP-NCAR and GDAS reanalyses.

## 3. Results

### 3.1 Asian dust plume characteristics

Based on the joint probability distribution of dust plume top height and motion speed (Fig.3), dust particles from the Taklamakan and Gobi Deserts are injected to similar heights above sea level, despite generally lower surface elevations across the Taklamakan Desert. Over both deserts, the injection heights are most frequently observed between 1000 to 1500

m ASL, i.e. from the surface to about 500 m above the ground. About 5% of dust plumes are injected to higher than 2 km ASL over both deserts. Taking into account the differential sensitivity of atmospheric suspension time to dust injection





height over the two deserts (Fig.2), dust emitted from the Taklamakan Desert appears to have higher potential for long-range transport than dust emitted from the Gobi Desert. According to the trajectory analysis, under dry conditions without wet deposition, dust particles injected to above 2 km ASL over the Taklamakan Desert often stay in the atmosphere for longer than 12 days, which requires an injection height above 2.5 km ASL from the Gobi Desert (Fig.2). In addition, the injection

height over both deserts appears to be independent of wind speed, similar with the findings based on MISR observations and a 1-D plume rise model regarding smoke plumes associated with wildfires (Sofiev et al., 2012). The elevated dust plumes are mainly associated with weak winds below 5 m s$^{-1}$ over the Taklamakan Desert, compared with strong winds exceeding 15 m s$^{-1}$ over the Gobi Desert. The different wind regimes associated with dust injection height are primarily due to soil characteristics across the two deserts, namely primarily fine sands across the Taklamakan Desert versus coarse, rocky soils

across the Gobi Desert (Sun et al., 2013).

Dust activation events occur most frequently in spring over both the Taklamakan and Gobi Deserts (Fig.4), consistent with previous findings based on the climatology of AOD from MODIS (Yu et al., 2008) and dust observations at weather stations in China and Mongolia (Lim and Chun, 2006), among other observational evidence. Over the Taklamakan Desert, the occurrence of dust activation peaks in April and remains active except for the boreal winter months of

November-February. An earlier study analyzing weather station, radiosonde, and reanalysis data suggests a key role of the nocturnal low-level jet on the dust activation over the Taklamakan Desert during spring and summer (Ge et al., 2016). Dust particles are injected to higher levels in boreal spring and summer, indicating seasonally enhanced potential for long-range transport of Taklamakan dust. Over the Gobi Desert, dust activation mostly occurs in March and April, driven by seasonally enhanced surface wind speeds associated with frequent cyclogenesis over northern China and Mongolia (Kurosaki and

Mikami, 2004). Unlike the year-round dryness across the Taklamakan Desert, the Gobi Desert is affected by the East Asian summer monsoon, which inhibits summertime dust activation across the Gobi Desert (Arimoto et al., 2006). Dust injection height also peaks in March and April over the Gobi Desert, suggesting the highest potential for long-range dust transport during these two months. The July peak of dust injection height over the Gobi Desert is likely due to enhanced ascent (Fig.S2) but needs further investigation, especially given the limited samples of dust plumes during July in the current study.

The climatology in horizontal wind, especially the zonal wind, is less favorable for eastward dust transport from the Taklamakan Desert than from the Gobi Desert (Fig.5). Around 2 km ASL, the predominant wind direction is easterly in most months over the Taklamakan Desert, compared with dominant westerlies over the Gobi Desert at most vertical levels in most months. However, stronger ascending motion is noted at most vertical levels in most months over the Taklamakan Desert than over the Gobi Desert (Fig.S2), including especially the climatological ascent from the surface to about 3 km ASL

during April to October; this indicates a higher potential for dust over the Taklamakan Desert to become elevated to greater heights and enter the mean westerlies above 4 km ASL. Indeed, vertical motion exerts a strong influence on the travel distance of dust particles, as indicated by the correlation between dust particle atmospheric suspension time and first-three-hour-average vertical motion (-0.61 for Taklamakan and -0.58 from Gobi, both p's < 0.01) according to 10,867,131 trajectories from the Taklamakan Desert and 310,290 trajectories from the Gobi Desert. In contrast, none of the correlations



between atmospheric suspension time and horizontal winds (U, V, or wind speed) are significant for either desert. The identified key role of vertical motion in determining the dust particle atmospheric suspension time confirms previous findings based on dust events during the ACE-Asia field campaign (Tsai et al., 2008).

### 3.2 Asian dust transport pathways

The dust transport pathways are complicated after emission from the Taklamakan Desert (Fig.6). During the first two days since emission, dust particles are mainly observed over limited areas in northern China and Mongolia near the source. Starting from the third day after emission, dust particles from the Taklamakan Desert begin to affect Korea, Japan, and the northwestern Pacific Ocean and reach the west coast of North America as early as on the sixth day since emission. By day 12 since emission, Taklamakan dust particles reach scattered areas of North America, but are more frequently present over other

parts of the Northern Hemisphere. Given the widespread nature of dust after long-range transport, the absolute values in the spatial distribution of trajectory endpoint are less informative than the spatial distribution pattern itself, especially after several days since emission. In summary, the trajectory passage suggests that dust emitted from the Taklamakan Deserts undergoes various, complicated routes, with a small portion reaching North America as early as on the sixth day since emission.

Dust from the Gobi Desert undergoes even more complicated transport pathways that are closer to the Arctic compared with dust from the Taklamakan Desert (Fig.7). Since the Gobi Desert is located to the east of the Taklamakan Desert, Gobi dust begins to influence the northwestern Pacific earlier than Taklamakan dust. However, due to complicated meteorological conditions, likely associated with polar fronts and jet streams, Gobi dust does not arrive to North America until day eight after emission. Compared with the spatial distribution of Taklamakan dust influence, Gobi dust exerts an

influence over wider areas, ranging from as far south as 10˚N over the Pacific Ocean to as far north as the Arctic.

### 3.3 Seasonality in Asian dust transport

The transport of Taklamakan dust exhibits substantial seasonal variability (Fig.8). The area affected by Taklamakan dust generally ranges from 30˚N to 60˚N, with greater influence in East Asia near its source in all months. Consistent with implications from injection height (Fig.4), Taklamakan dust is frequently transported over a long distance in both spring and

summer. Along with the seasonal placement of the predominant mid-latitude westerlies, the spread of Taklamakan dust exhibits a northward shift from spring to summer, mainly affecting 30˚N-45˚N over North America in spring (April-May) and 45˚N-60˚N in summer (June-August). In May, both polar and subtropical jet streams contribute to the long-range transport of Taklamakan dust, as suggested by the apparent bifurcating transport paths (Fig.7b). Moreover, in May, 1-5% of trajectories from the Taklamakan Desert carry dust to the south and southeast Asia, exerting potential influence on the onset

of South Asian summer monsoon. The presence of Taklamakan dust over south and southeast Asia, according to observation-initiated trajectories, supports the findings regarding the influence of Asian dust on the South Asian summer monsoon by previous modeling study (Lau et al., 2006). In July, the troughing pattern of trajectories east of Japan is likely



caused by the frequent occurrence of extratropical low pressure centers associated with the sea-surface temperature gradient across the Kuroshio current and its extension, along with the subtropical high pressure well developed in summer centered around 30˚N, 170˚E (Tanimoto et al., 2011).

Substantial seasonality is also present in the trajectory passages of dust particles emitted from the Gobi Desert (Fig.9). Similar to dust particles from the Taklamakan Desert, the springtime dust from the Gobi Desert generally spreads wider than summertime dust, consistent with the implication from the seasonal cycle of injection height (Fig.4). The constrained and northward spread of Gobi dust in June is likely due to seasonally weakened and northward displacement of polar jet streams. In March, 1-5% of dust trajectories with initial injection height above 2 km ASL from the Gobi Desert travel over the Bering strait to the Arctic, likely causing phenomena like "Arctic haze" at several kilometers above the ground (Rahn et al., 1977).

## 3.4 Spread of Asian dust to North America

Taklamakan dust generally exerts a geographically wider spread over North America, especially over the southern part of the continent, than Gobi dust, according to trajectory analysis (Fig.10). Overall, 5756 (3.9%) dust trajectories from the Taklamakan Desert and 3804 (3.7%) dust trajectories from the Gobi Desert travel over North America each year during 2001-2003, affecting 31% and 23% of the total area by more than 100 dust trajectories per year from the Taklamakan and Gobi Deserts, respectively. Both the number of trajectories passing through and the area affected by a substantial number of trajectories from the Taklamakan Desert maximize around 40˚N over North America. In contrast, both statistics maximize around 50˚N-55˚N over North America for Gobi dust trajectories. Around 40˚N over North America, about 500 trajectories from the Taklamakan Desert and 150 trajectories from the Gobi Desert are observed each year during 2001-2003, with about 60% and 20% of the North American area affected by more than 100 trajectories per year from the Taklamakan and Gobi Deserts, respectively. Around 55˚N, these numbers are 300 and 400 trajectories per year from the Taklamakan and Gobi Deserts, respectively, and 10% and 20% of the area affected by more than 100 trajectories per year from the Taklamakan and Gobi Deserts, respectively. The latitudinal distribution of the influence from both deserts are likely due to the relatively strong northerly wind over the Taklamakan Desert compared with the Gobi Desert (Fig.5). The relatively wider spread of influence of Taklamakan dust, compared with Gobi dust, on North America is consistent with the implications from the dust injection height (Fig.3), atmospheric suspension time (Fig.2), and local meteorological conditions (Fig.5 and Fig.S2) regarding the higher potential of long-range transport from the Taklamakan Desert. The latitudinal distribution in the influence of Taklamakan dust on North America has been verified based on additional trajectories during 2006-2011 using both NCEP-NCAR and GDAS1 as the meteorological drivers, which also exhibits a peak between 40˚N-45˚N (Fig.S4).

## 3.5 Interannual variability in Asian dust activation and transport

Activation occurrence, injection height, and long-range transport of dust emitted from the Taklamakan Desert exhibit substantial interannual variability during 2001-2011 (Fig.11). The number of dust activation events captured by MISR and

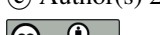



retrieved by MINX varies from 38 events in 2003 to 127 events to 2004. The inhibited dust activation in 2003 appears to be associated with anomalous wet soils, which represent the wettest year in an land surface model ensemble simulation of regional soil moisture in China during 1950-2006 (Wang et al., 2011). The median dust injection height varies from 1334 m ASL to 1640 m ASL in 2003, suggesting substantial variability in the potential for long-range transport. In addition, the

transport patterns vary by year, as indicated by the influence on the remote regions such North America by latitude. 2004 appears to be the year with the most favorable meteorological conditions for trans-Pacific transport of Taklamakan dust, with the percentage of trajectories passing over North America maximizing at 10% over around 53˚N, likely attributed to two factors. One cause is the inhibited springtime dust activation (23 events in March-May) and enhanced summertime dust activation (89 events in June-August), with the later favoring a northward shift of the dust spread (Fig.8). Another factor is

likely the large-scale circulation associated with the El Niño starting from June that favors a northward shift of the Asian dust spread (Gong et al., 2006). In 2002 and 2011, the influence of Taklamakan dust maximize at a lower latitude of around 40˚N and 45˚N, respectively. In contrast, during 2006 and 2009, the spread of Taklamakan dust over North America is trivial.

Indeed, such interannual variability is also present in the activation and transport of dust emitted from the Gobi

Desert, as suggested by the three years of data (Fig.10). Limited by the short record for dust plumes over the Gobi Desert, we did not make an effort to examine interannual variability in dust emission and transport from the Gobi Desert. However, comparing the interannual variability in the influence on North America by the dust trajectories from both deserts during 2001-2003, it appears that the interannual variability in dust emission and transport is greater for Gobi dust than for Taklamakan dust. The greater interannual variability in Gobi dust emission and transport is potentially caused by the

complicated meteorological conditions, especially associated with the frontal activity (Tsai et al., 2008), over the Gobi Desert and along the Gobi dust transport pathways (Fig. 7 and 9).

## 4. Discussion and Conclusions

In the current study, the climatology in dust plume characteristics and long-range transport of dust from the Taklamakan and Gobi Deserts are examined using the climatology of dust heights and winds near the source derived from MISR stereo

observations of optically thick dust plume and trajectory analysis. Based on the unique MISR dust injection height and plume motion observations and confirmed with trajectory analysis, dust particles emitted from the Taklamakan Desert generally present higher potential for long-range transport and exert wider spread in their cone of influence across North America through trans-Pacific transport than dust from the Gobi Desert. The higher transport potential of Taklamakan dust is primarily attributed to greater injection heights from the ground, enabled by finer dust particles emitted from the Taklamakan

Desert and climatological mid-to-low level ascending motion in spring and summer. Furthermore, the relative abundance of Taklamakan and Gobi dust over North America depends on latitude, with greater influence from Taklamakan dust to the south of 50˚N and greater influence from Gobi dust to the north of 50˚N across North America, based on trajectory analysis.



Consistent with previous observational evidence, the MISR dust plume data and trajectory analysis indicate maximized dust activation occurrence and highest potential for long-range transport in spring from both deserts, with about 5% of the trajectories passing over North America. Substantial summertime dust activation and long-range transport from the Taklamakan Desert occur as well.

Uncertainty in the current results mainly comes from the assumptions in trajectory modeling. By treating dust particles as tracer in the trajectory modeling, processes such as wet deposition and gravitational settling are ignored. Therefore, the actual frequency of long-range dust transport, in particular the trans-Pacific dust transport from Asian sources to North America, is overestimated in the current trajectory analysis. Taking gravitational settling into account, the actual potential for long-range transport of dust from the Taklamakan Desert, compared with the Gobi Desert, is probability even

higher than assessed in the current study, given the smaller particle size of Taklamakan dust. This hypothesis can be tested by analyzing particle size distribution along trajectories from both deserts using ground and satellite observations.

Furthermore, the accuracy of trajectory passages presented in this study is limited by the usage of the coarse-resolution NCEP-NCAR reanalysis as meteorological driver. Although the conclusions regarding the sensitivity of atmospheric suspension time to dust injection height and influence of Asian dust on North America have been confirmed

with trajectory analysis driven by a newer reanalysis with higher spatial resolution (Fig.S3 and Fig.S4), the trajectory analysis-based findings regarding Asian dust transport and relative contribution of Taklamakan and Gobi dust on remote regions such as North America need to be confirmed by future observational and modeling efforts. As an extension of previous modeling studies (Chen et al., 2017; Zhang et al., 2003), model simulations of the size-dependent dust emission from the Taklamakan and Gobi Deserts are encouraged to incorporate observational constraints provided by MISR aerosol

and particle shape measurements at different wavelengths as well as multi-spectral measurements from other satellite instruments (Wang et al., 2012; Xu et al., 2017). In order to verify the identified seasonality in dust transport patterns, we suggest future studies to take advantage of both geostationary and polar-orbiting satellite observations, as well as ground-based lidar observations. In order to confirm our findings about the latitudinal distribution of the influence from Gobi and Taklamakan dust, we encourage future studies to examine isotopic abundance, mineralogical composition, and particle color

information from samples collected at different locations across North America.

The present results on the interannual variability in dust activation and transport motivates further observational investigations on the natural and anthropogenic drivers of such variability. The aforementioned satellite observations, especially the long-term stereo and optical measurements from MISR, will facilitate such observational investigation of the interannual variability in dust emission and transport from Asia, as demonstrated by previous examples focusing on North

Africa and Middle East (Notaro et al., 2015; Yu et al., 2018). Future observational studies on the environmental drivers of Asian dust emission and transport will also benefit from the development of advanced statistical methods, such as the Stepwise Generalized Equilibrium Feedback Assessment which has been successfully applied to examine the environmental drivers of North African dust and climate variability (Yu et al., 2017a, 2017b).



## Acknowledgements

MISR CMVP data was obtained from the NASA Langley Research Center Atmospheric Science Data Center. MERRA-2 meteorological reanalysis was obtained from Goddard Earth Sciences Data and Information Services Center (GES DISC). The MINX dust plume data was processed by Mr. Michael Goetz during his 2012 summer internship at JPL and stored at the Jet Propulsion Laboratory. Dr. Yu was partially funded by NASA's Postdoc Program (NPP). This work was performed at the Jet Propulsion Laboratory, California Institute of Technology, under a contract with the National Aeronautics and Space Administration. The authors thank the MISR team for providing facilities and useful discussions.

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





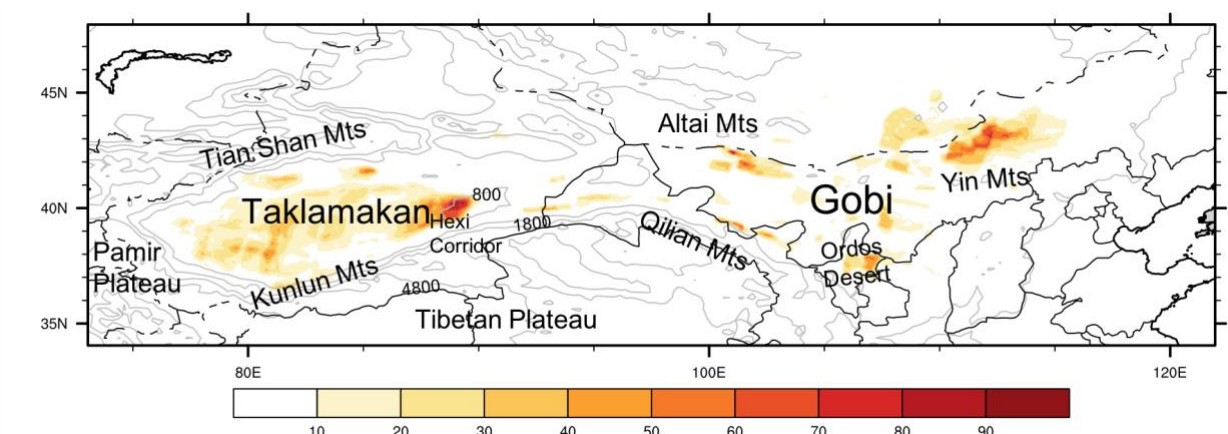

**Figure 1: Spatial distribution of dust plume detection frequency (%sample maximum) according to MINX. Grey contours indicate surface elevation (m) from the MISR Digital Elevation Model (DEM). Dust plume detection frequency in each pixel is calculated as the number of detected dust plume points per year.**

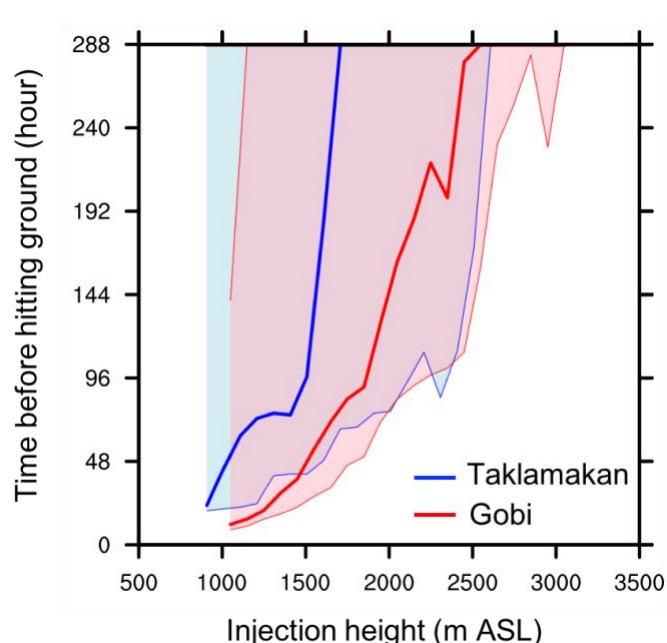

**Figure 2: Atmospheric suspension time (hours) of dust particles emitted from the Taklamakan (40˚N, 89˚E, elevation = 805 m) (blue) and Gobi Deserts (43.5˚N, 130˚E, elevation = 954 m) (red) as a function of injection height (m ASL), based on trajectories in March-May of 2001-2003. The thick lines (shading) represent the median (10th to 90th percentiles) of suspension time among 276 trajectories for each injection height.**



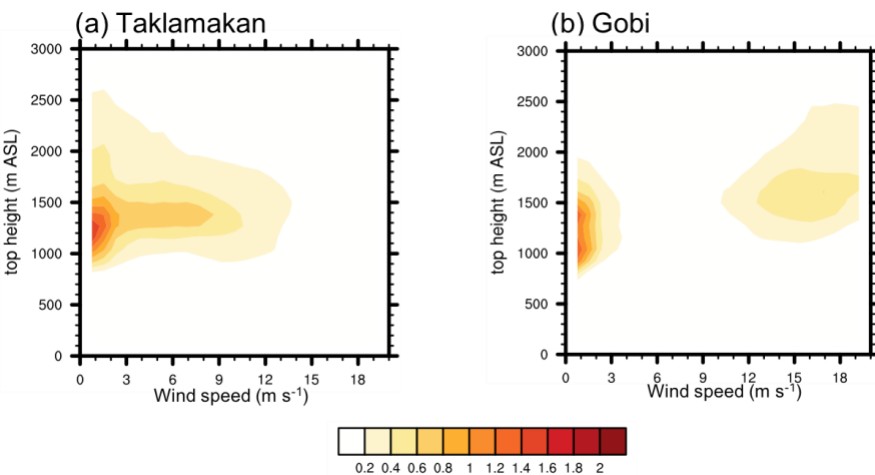

**Figure 3: Joint Probability Density Function (%) of dust plume top height (m ASL) and moving speed (m s⁻¹) across the (a) Taklamakan Desert during 2001-2011 and (b) central Gobi Desert during 2001-2003 from MINX dust plume observation.**

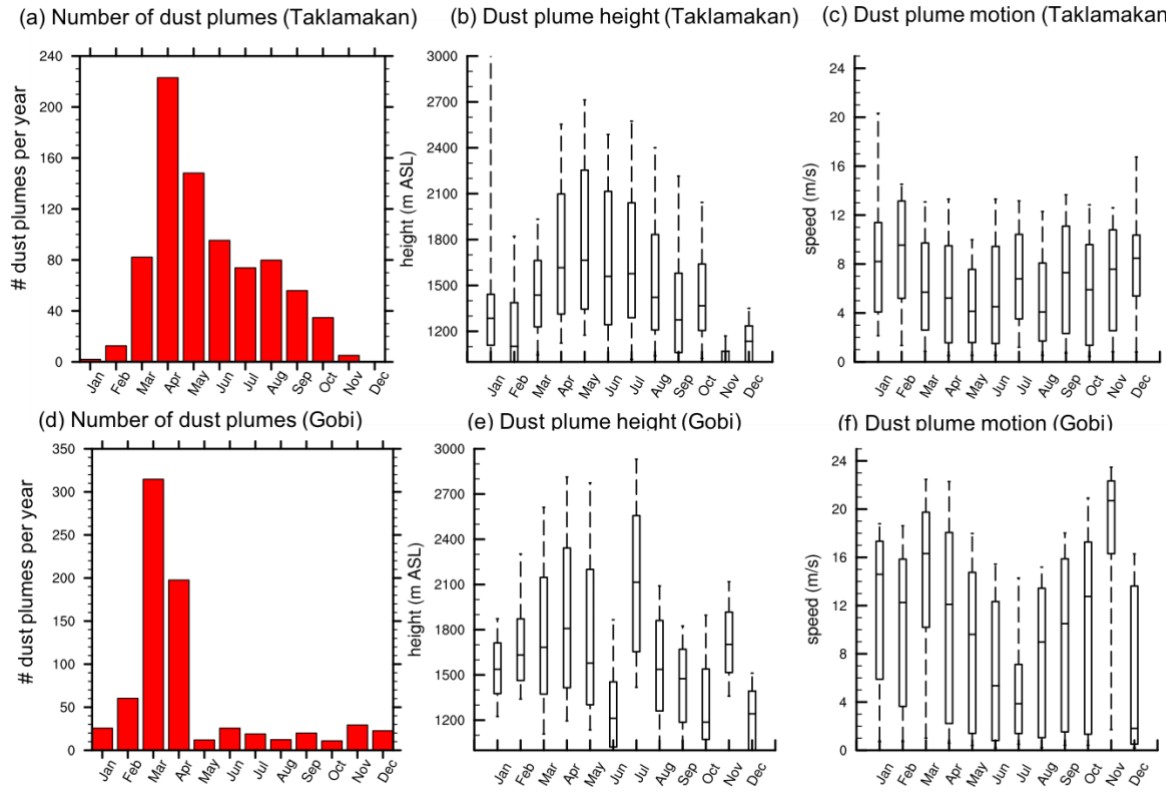

**Figure 4: Seasonality in dust plume characteristics from MINX dust plume data. (a, d) Dust plume occurrence per year bar chart, (b, e) injection height boxplot (10th, 25th, 50th, 75th, and 90th percentiles), and (c, f) plume top moving speed boxplot, across the (a-c) Taklamakan (2001-2011) and (d-f) Gobi (2001-2003) Deserts, respectively, by month.**





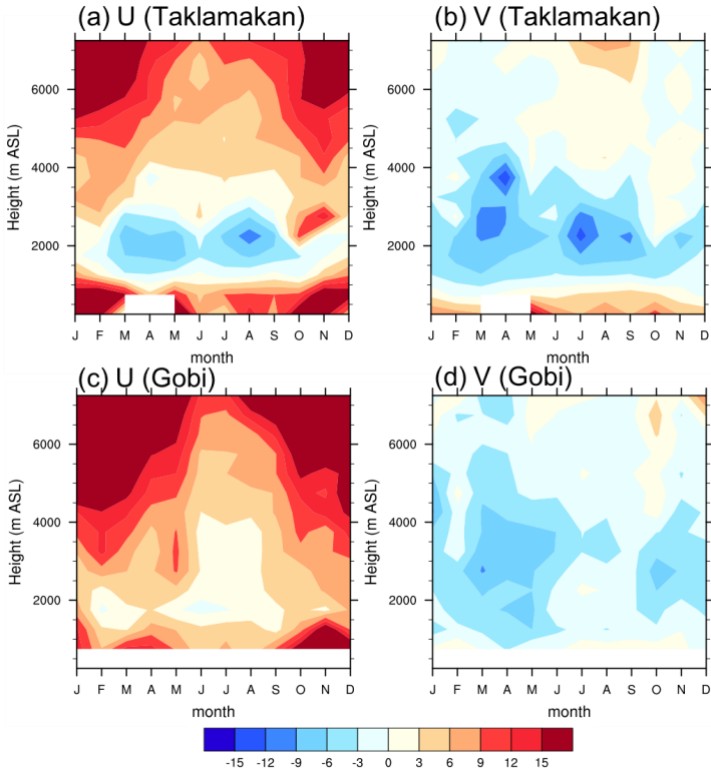

**Figure 5: Regional average (a, c) zonal (U) and (b, d) meridional (V) wind (m s-1) climatology across the (a-b) Taklamakan and (c-d) Gobi Deserts, by month and height. The wind data is from MISR CMVP during 2001-2017, sampling the days with optically thick dust plumes or clouds. The missing cells represent levels below ground or sampled by CMPV for less than 10 times.**




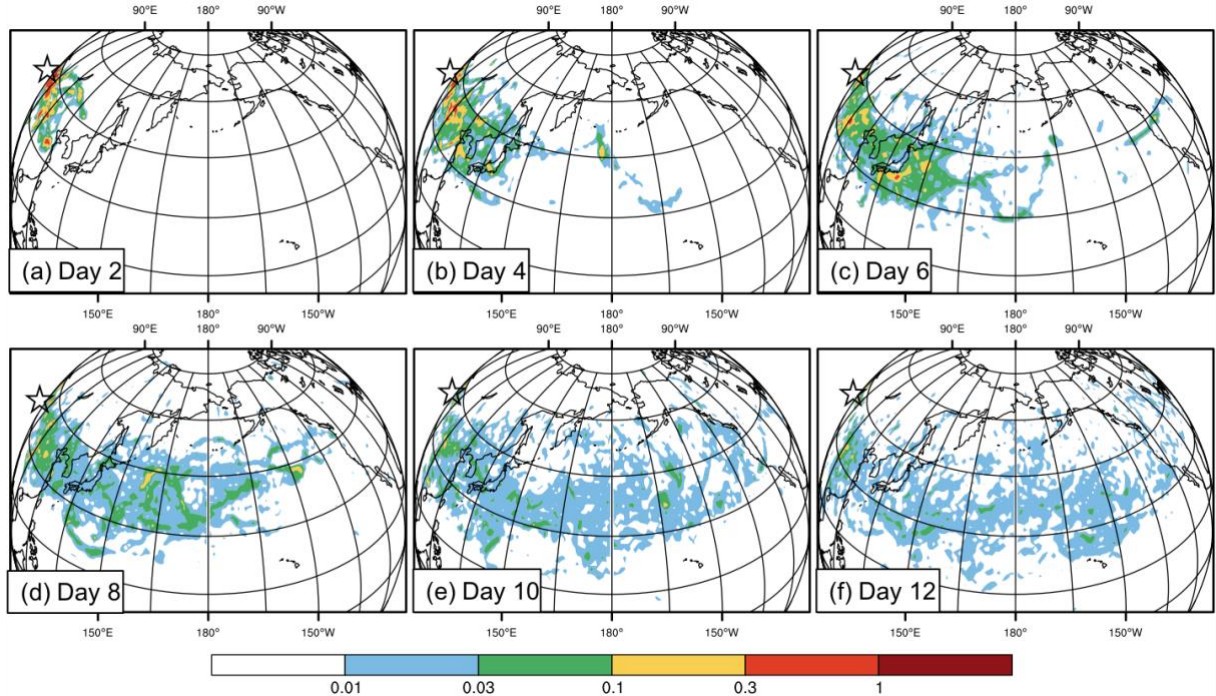

**Figure 6: Spatial distribution (%) of trajectory endpoints by the end of day (a) 2, (b) 4, (c) 6, (d) 8, (e) 10, and (f) 12 since emission from the Taklamakan Desert during 2001-2011. The spatial distribution values over all 1° latitude x 1° longitude grids in the Northern Hemisphere sum up to 100%. The star indicates the center of all emission points.**





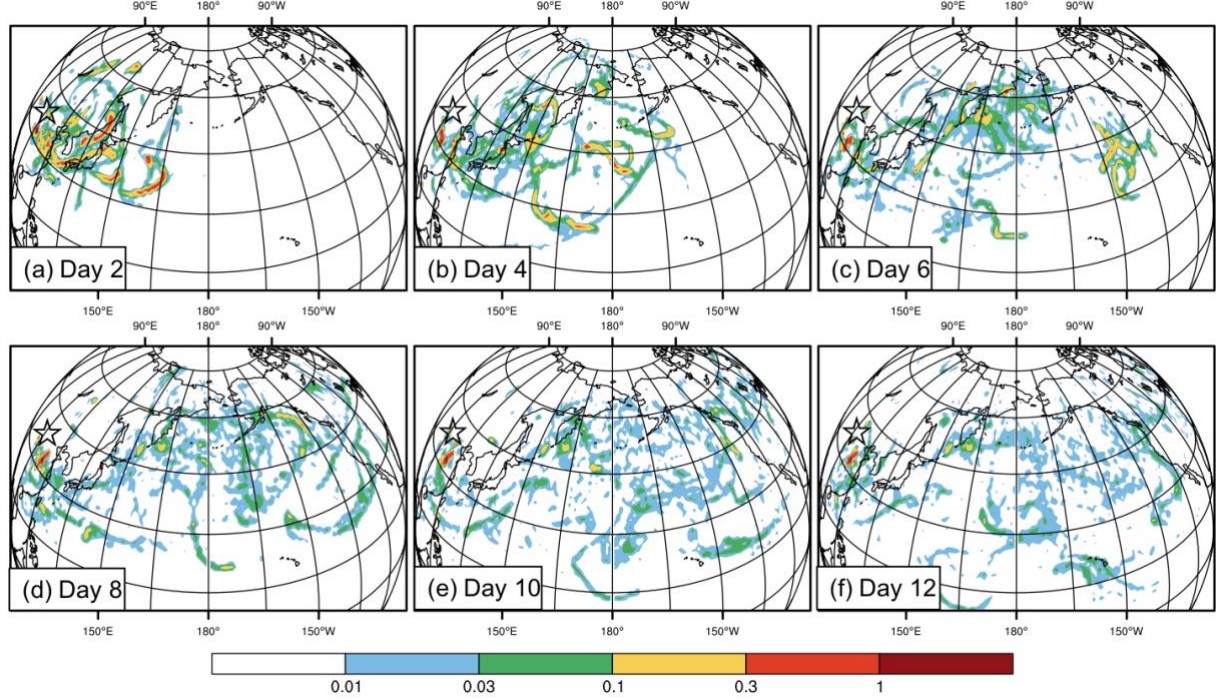

**Figure 7: Spatial distribution (%) of trajectory endpoints by the end of day (a) 2, (b) 4, (c) 6, (d) 8, (e) 10, and (f) 12 since emission from the central Gobi Desert during 2001-2003. The star indicates the center of all emission points.**





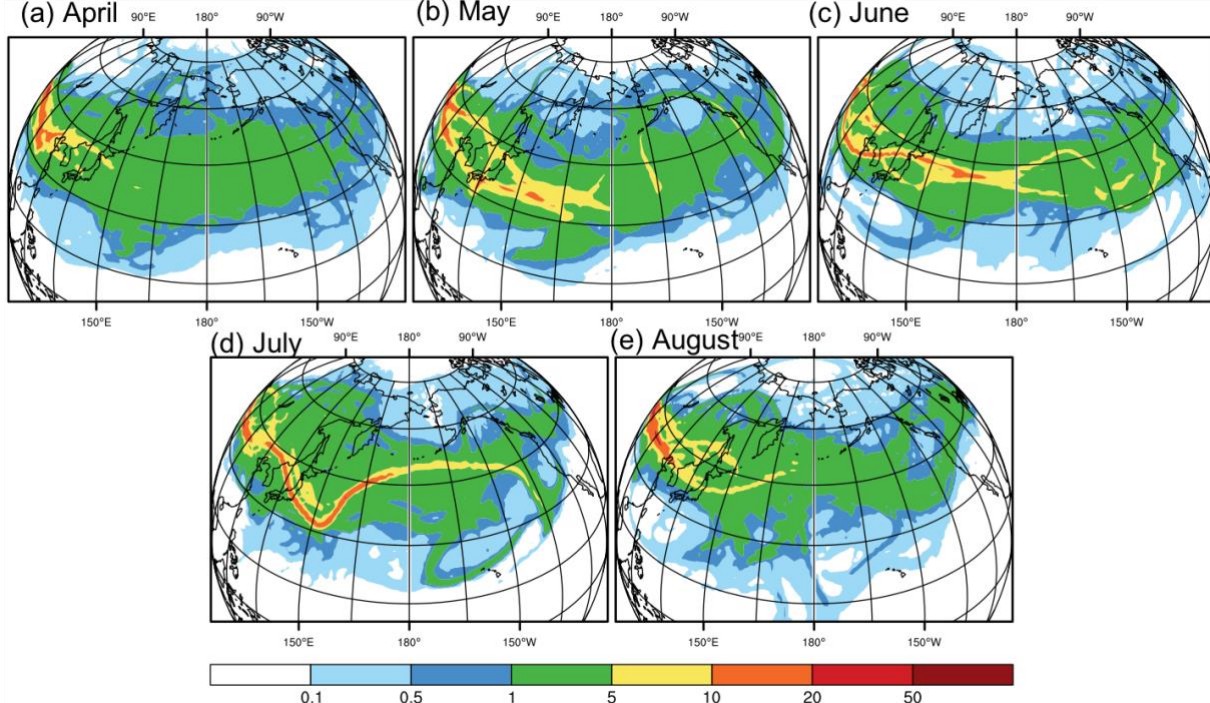

**Figure 8: Trajectory passage frequency (% of trajectories) from the Taklamakan Desert in (a) April, (b) May, (c) June, (d) July, and (e) August during 2001-2011. These months are analyzed because there are at least 10,000 data points with initial injection height exceeding 2 km ASL in each of these months. The trajectory passage frequency is calculated as the number of trajectories crossing over each 1° latitude x 1° longitude grid in the Northern Hemisphere divided by the total number of trajectories from the Taklamakan Desert in each month.**

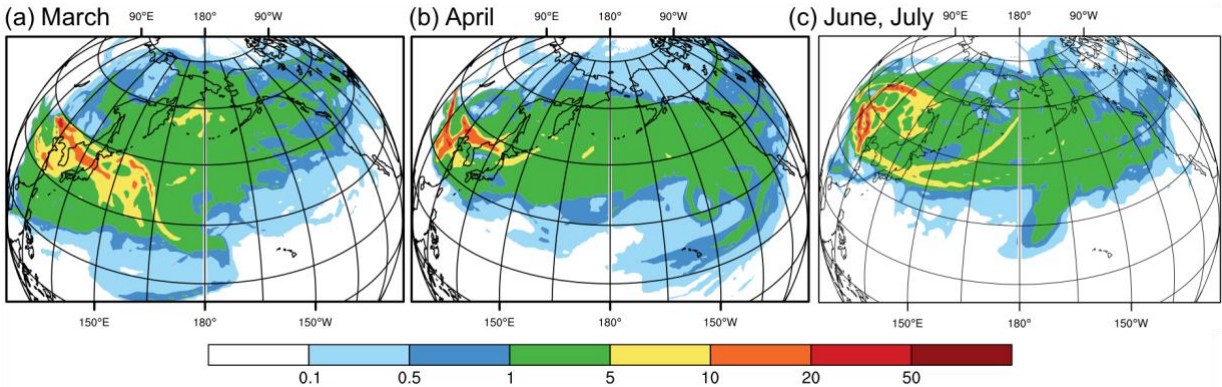

**Figure 9: Trajectory passage frequency (% of trajectories) from the central Gobi Desert in (a) March, (b) April, and (c) June and July during 2001-2003. There are at least 2,500 data points with initial injection height exceeding 2 km ASL in each of the analyzed periods. The trajectory passage frequency is calculated as the number of trajectories crossing over each 1° latitude x 1° longitude grid in the Northern Hemisphere divided by the total number of trajectories from the Gobi Desert in each month.**



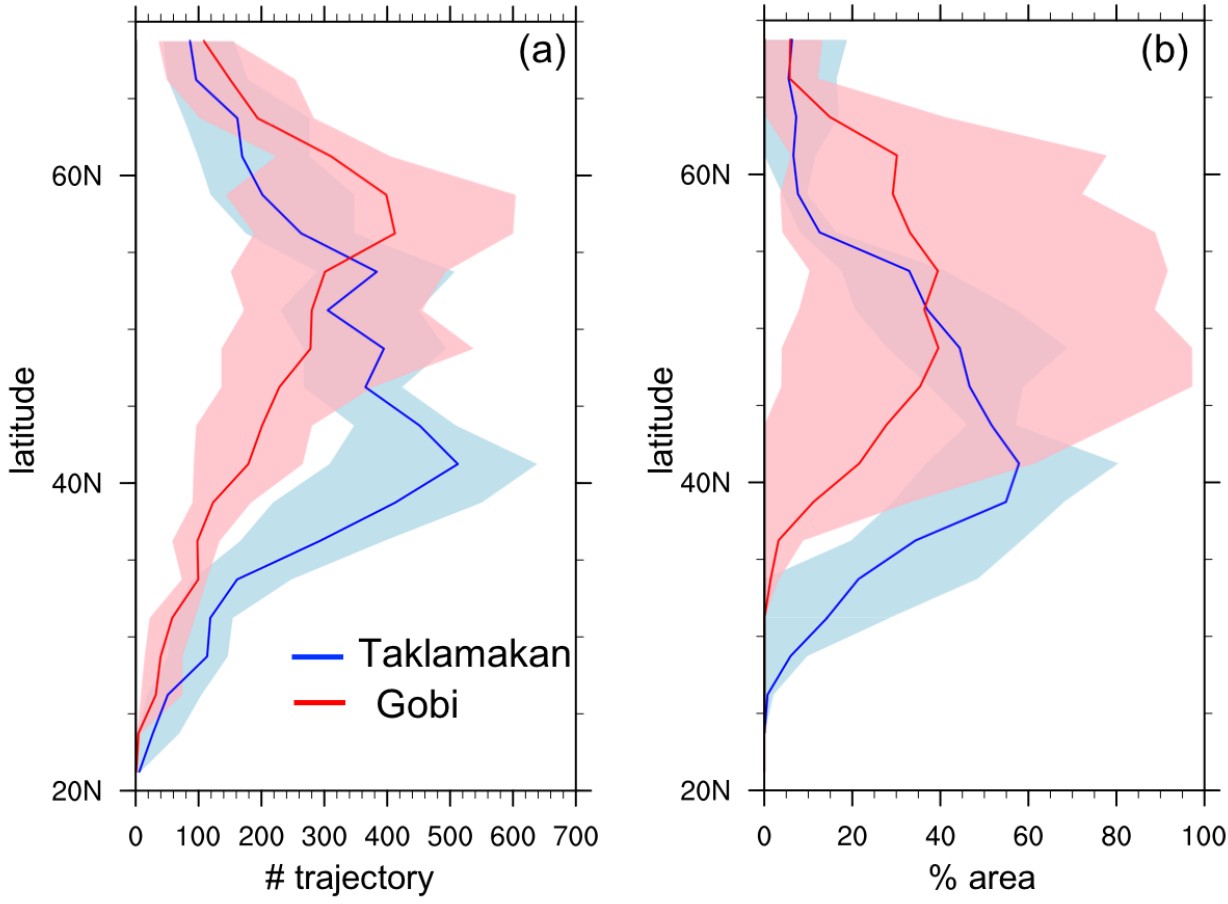

**Figure 10: Spread of Taklamakan and Gobi dust to North America by latitude, represented by trajectory passage during 2001-2003. (a) Number of trajectories per year from Taklamakan (blue) and Gobi (red) that pass over each 2.5° latitude band. (b) Percentage of area in each 2.5° latitude band influenced by more than 100 trajectories per year from Taklamakan (blue) and Gobi (red). The shadings represent maximum and minimum values among 2001, 2002, and 2003. These years are selected for the analysis because MISR dust plume data are available for both deserts during that period.**





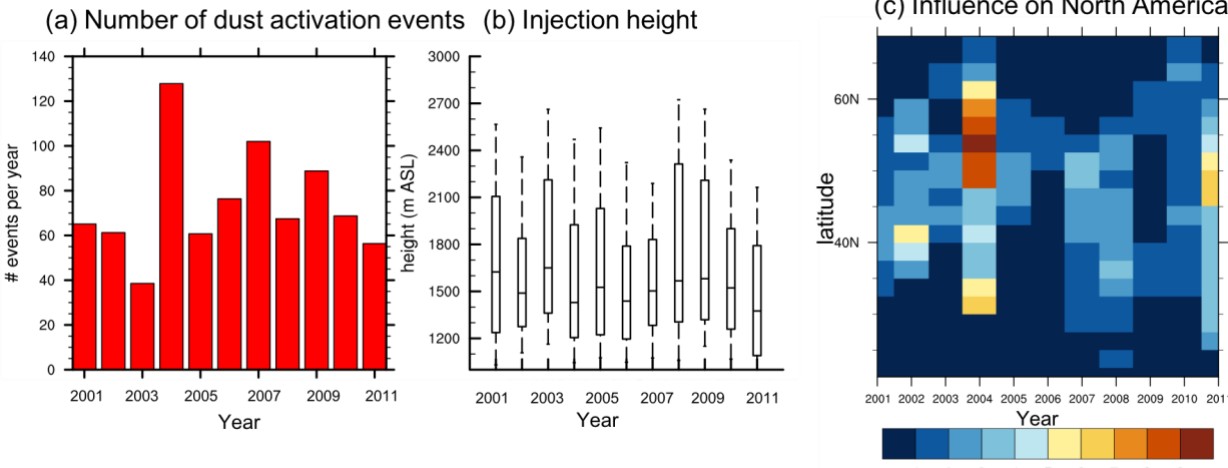

**Figure 11: Interannual variability in Taklamakan dust activation and transport during 2001-2011. (a) Dust activation occurrence, (b) injection height boxplot (10th, 25th, 50th, 75th, and 90th percentiles), and (c) percentage of trajectories passing over North America by latitude.**