# Peer review of "Climatology of Asian dust activation and transport potential based on MISR satellite observations and trajectory analysis"

_Atmospheric Chemistry and Physics, 2018_

## Referee Comment (RC1) · Anonymous Referee #1 · 2 Oct 2018

**Review comment on "Climatology in Asian dust activation and transport based on MISR satellite observations and trajectory analysis" (https://doi.org/10.5194/acp-2018-857)**

**General comments:**

Yu et al. describe long-range transport of Asian dust with satellite observations and forward trajectories. The MINX tool was used to retrieve dust injection heights at the Gobi and Taklamakan deserts from measurements of the MISR instrument on board the Terra satellite. The obtained heights were consequently used to initiate HYSPLIT forward-trajectories, which were then used to describe the trans-Pacific dust transport and its seasonal and spatial variation. Such long-term data sets are useful for statistical analyses of dust layer heights and dust transport, needed for radiative-transfer calculations and aerosol-cloud interaction studies. This is of high importance, especially for regions with sparse coverage of ground-based observations.

The manuscript is well written and I suggest publication in ACP after clarification of a few points (see specific comments) and some minor technical corrections.

**Specific comments:**

Concerning the temporal resolution of the MISR observations (crossing of the study region every 6-7 days), I have some questions about the dust events and plume data points: For example for the 2-3 years of analyzed data in the Gobi desert: Let us assume, 3 years, 1095 days. Now we have an observation every 6.5 days, so we have 168 observations. At 2251 dust events, this means 13 dust events per observation. How do you separate those dust events, source regions (intra-desert), and finally injection heights?

Concerning injection heights: You state the method works independently of background aerosol and thin cirrus, but how do you deal with complex multilayer structures of (possibly) optically very dense dust layers?

Concerning background aerosol: Do you consider or observe high-altitude layers of (polluted) dust probably of western origin as described for example in Tanaka et al., 2005 (https://doi.org/10.1016/j.atmosenv.2005.03.034), Mikami et al., 2006 (https://doi.org/10.1016/j.gloplacha.2006.03.001), and Hofer et al., 2017 (https://doi.org/10.5194/acp-17-14559-2017)

Concerning the difference in particle sizes of Gobi and Taklamakan dust: You mention this difference for soil (Sun et al., 2013), and that it probably has implications on the potential far range transport of Gobi dust compared to Taklamakan dust. However, what about the actual mobilization and air-borne dust if coarser particles were mobilized, as you state, at higher winds in Gobi? It could compensate to a certain degree the gravitational settling (for example, Gasteiger et al., 2017 (https://www.atmos-chem-phys.net/17/297/2017/) try to explain the long-range transport (westward, though) of very coarse dust particles).

**Technical corrections (minor formatting and typing errors):**

Page 2 Line 18: In general, locked space between Fig. and number.

Page 2 Line 20: Tian Shan Mountain -> Tian Shan Mountains or Tian Shan Mountain system

Page 4 Line 27/26: leave "elevation" out or at least write ASL behind the values.

Page 6 Line 8: 15 m (line break) s^-1 -> use locked spaces ~ before (and within) units

Page 6 Line 34: p's -> p-values

Page 9 Line 2: "the wettest year in an land surface model ensemble simulation" -> "an" is wrong here, "any" or "a"?

Page 10 Line 34: Yu et al., 2017a, 2017b -> Yu et al. 2017a,b

Page 11 Line 21 : "Geosci. Remote Sensing, IEEE Trans." -> Abbreviation and comma are wrong, use "IEEE Trans. Geosci. Remote Sens."

Page 11 Line 29: add number (16) to volume 8. Furthermore, the doi is only here formatted as a web link, I think this is nowadays standard at ACP, use it everywhere.

Page 12 Line 15: Pages are wrong. The article has a page-like number (L06824), it might need to be stated, I don't know, but it has to be consistent within the References.

Page 12 Line 18: The article has a page-like number (L19802).

Page 12 Line 21/22: The article has a page-like number (D23212).

Page 12 Line 27: doi is wrong, digit missing and with a space. The correct one is 10.1002/2014JD021796. Check capitalization in the title, "East Asian".

Page 12 Line 32: The article has a page-like number (114018)

Page 12 Line 33: Rest of the authors is missing. Huebert, Bates, Russel, etc.

Page 13 Line 3: Here it is not "J. Geophys. Res. Atmos.", it is only "J. Geophys. Res.". Furthermore is the number D20 and the page-like number 9000.

Page 13 Line 4/6: "Kahn, R. a." -> "Kahn, R. A.". Probably this conference contribution is cited better like this (as it is stated on the SPIE homepage): "Proc. SPIE 8177, Remote Sensing of Clouds and the Atmosphere XVI, 81770O (26 October 2011)" plus doi and year, of course. Check the page numbers.

Page 13 Line 8: The article has a page-like number (L03106)

Page 12 Line 21: Why "(April 2006)"?

Page 12 Line 32/33: This reference is insufficient. Do you mean this? https://eospso.gsfc.nasa.gov/sites/default/files/atbd/MISR_L3_CMV_ATBD.pdf Add the link and date of last access.

Page 14 Line 4: Journal is missing, it is: "Proc. SPIE 7089, Remote Sensing of Fire: Science and Application, 708909 (27 August 2008)"

Page 14 Line 5:  "Kahn, R. a. and Dunst, B. a." -> "Kahn, R. A. and Dunst, B. A."

Page 14 Line 11: Take care of the special characters and capitalization.  ACP provides a working bib file: AUTHOR = {Salvador, P. and Alonso-P\'erez, S. and Pey, J. and Art\'{\i}\~nano, B. and de Bustos, J. J. and Alastuey, A. and Querol, X.},

Page 14 Line 16/17: Check capitalization. The title is: NOAA's HYSPLIT Atmospheric Transport and Dispersion Modeling System.

Page 14 Line 20: The article has a page-like number (D05207). Again „Atmos." is not necessary in the older JGR articles (I think before 2013), maybe even wrong, I am confused. Either you put „Atmos." everywhere (each issue D = Atmospheres) or nowhere in these articles.

Page 14 Line 26: "Atmos."? And in that specific article there are commas for digit grouping (strange, I know): 10,325-10,333. The same is in Huang et al., 2014, but there it is already correct. If you want to leave it out, you have to leave it out there as well.

Page 14 Line 29/30: Journal abbreviation is wrong, it should be "Geochem. Geophys. Geosyst."

Page 15 Line 4/5: "Atmos."?  and the article has a page-like number (D17311)

Page 15 Line 9: "Atmos."?  and the article has a page-like number (D12213).

Page 15 Line 11: "theglobe" -> "the globe".

Page 15 Line 23: "Tsay S. C." needs hyphen "Tsai S.-C." like in Wang et al., 2012b and the article has a page-like number (L08802).

Page 15 Line 27: Why (X)? Maybe (Part A).

Page 15 Line 29: Replace "n/a-n/a". The article has a page-like number (L05811).

Page 15 Line 33: "Atmos."?  and the article has a page-like number (D00H35).

Page 16 Line 1: Why (April 1998)?

Page 16 Line 4: Here "Atmos." would fit.

Page 16 Line 6: (May) probably not needed, pages neither, article number is 15333. Like in Yu et al. 2017a.

Page 16 Line 21: No pages, the article has a page-like number (L07603).

Page 16 Line 23: No pages, the article has a page-like number (L18815).

Page 16 Line 25: No pages, the article has a page-like number (2272).

Page 17 Figure 1: "(%sample maximum)" -> "(% sample maximum)" or "(% of sample maximum)"

Page 17 Figure 2: Add ASL behind elevation values.

Page 18 Figure 4: The subfigures d) e) f) slightly overlap their titles. In general, the figures need a bit a higher dpi for final publication.

Page 24 Figure 11: Caption, 10th, 25th etc. not superscript as in the caption of Fig. 2 and 4. Please be consistent.

Supplement Figure S3: Add ASL behind elevation values. And full stop or colon behind bold figure numbers in general.

---

## Referee Comment (RC2) · Anonymous Referee #2 · 25 Oct 2018

The paper "Climatology in Asian dust activation and transport based on MISR satellite observations and trajectory analysis" presents and discusses the transport of dust aerosols, emitted from the arid and semiarid deserts of Taklamakan and Gobi, over the northern Pacific Ocean. The study falls within the scope of ACP. The study is based on MISR observations, MINX aerosol top height, and accordingly, forward HYSPLIT trajectory analysis. The manuscript is well-written/structured, the presentation clear, the language fluent. However, the submitted study is subject to major deficiencies and I would recommend publishing in ACP considering major revision.

Comments:

1) Regarding the "Asian dust activation climatology". Dust aerosol classification is crucial in the scope of the study, since it is the initial point of the trajectories analysis. Therefore, I would recommend to the authors to describe briefly the dust aerosol classification in MISR/MINX (including the necessary references). The scientific methods and assumptions are not clearly outlined. How is a "dust plume" defined in the paper and how is a "dust event"? In addition, in case of air parcels containing dust aerosols originating from both the Taklimakan and Gobi desert, how is the discrimination performed to the different sources? Which are the uncertainties in the classification?

2) Regarding the "Asian dust transport climatology". Although the paper presents an interesting approach to study dust transport the results are not sufficient to support the conclusions, due to the lack of observations provided on parallel with the trajectories. The study uses MISR observations-MINX provided top height to initiate HYSPLIT forward trajectories. Accordingly the climatology of trajectories is provided and not the Asian dust transport climatology. The difference is substantial. HYSPLIT computes the air parcel's transport and dispersion from a source region (Taklamakan and Gobi here) and describes where the air parcel will go. In the framework of the study, the climatology of the trajectories is provided (spatial distribution - % of trajectory endpoints / Trajectory passage frequency - % of trajectories after a specific number of days), without providing any observation/evidence on the presence of dust (per trajectory, distance or area). Dust aerosols may already have been removed along the transport/trajectory due to dry (gravitational settling) or wet deposition, although the air parcel will reach further distances. The paper does not even provide quantitative information on the probability of dust to have been transported. The trajectory may extend over the Pacific Ocean, and even further, to the western coast of the United States, however this does not provide any guarantee that dust is present and has reached that distance. I would suggest the authors to do any necessary modifications to the manuscript. Either provide dust observations per trajectory or to focus on the trajectories analysis without giving the impression on the presence (and transport) of dust to the trajectories endpoint. Which are the uncertainties? Alternatively, the authors could implement observations on the presence of dust to the western coast of USA (i.e. AERONET and AE, MODIS DT AOD and AE over ocean/ CALIOP volume/particle depolarization ratio) and use HYSPLIT back-trajectories. In addition, assuming a dust plume over an area, HYSPLIT initiated at different altitudes may provide different dust transport pathways. Therefore the study is representative only for the trajectories of the dust top-height and not for the dust plume (trajectories initiated at center of mass/scale height most probably would be more representative to discuss dust transport climatology).

3) The title does not reflect the contents of the paper and is misleading.

4) The figures are of high quality. I would suggest on Figure 2 to reverse the axes, time on horizontal axis and Injection height at the vertical height.

5) Regarding Figure 3 and the Gobi Desert the lack of continuity in wind speeds between 3 and 12 m/s is a strange feature. I would suggest the authors to describe this feature.

6) Regarding references, a brief list of references is provided. I would suggest the authors to expand the list of references in order to strengthen the manuscript and at the same time to give credit to related work. Indicatively, here a brief list of related studies is provided, describing features of dust aerosol transport emitted from the Taklamakan and Gobi deserts, based on synergies of passive and active ground-based and satellite-based instrumentation, models, campaigns and the meteorological and topographical mechanisms.

Bory, A. J. M., Biscaye, P. E. and Grousset, F. E.: Two distinct seasonal Asian source regions for mineral dust deposited in Greenland (NorthGRIP), Geophys. Res. Lett., 30(4), 1167, doi:10.1029/2002GL016446, 2003.

Chen, S., Huang, J., Li, J., Jia, R., Jiang, N., Kang, L., Ma, X. and Xie, T.: Comparison of dust emissions, transport, and deposition between the Taklimakan Desert and Gobi Desert from 2007 to 2011, Sci. China-Earth Sci., 60(7), 1338–1355,

doi:10.1007/s11430-016-9051-0, 2017.

de Leeuw, G., Sogacheva, L., Rodriguez, E., Kourtidis, K., Georgoulias, A. K., Alexandri, G., Amiridis, V., Proestakis, E., Marinou, E., Xue, Y. and van der A, R.: Two decades of satellite observations of AOD over mainland China using ATSR-2, AATSR and MODIS/Terra: data set evaluation and large-scale patterns, Atmos. Chem. Phys., 18(3), 1573–1592, doi:10.5194/acp-18-1573-2018, 2018.

Duce, R., Unni, C., Ray, B., Prospero, J. and Merrill, J.: Long-Range Atmospheric Transport of Soil Dust from Asia to the Tropical North Pacific - Temporal Variability, Science, 209(4464), 1522–1524, doi:10.1126/science.209.4464.1522, 1980.

Huebert, B. J., Bates, T., Russell, P. B., Shi, G. Y., Kim, Y. J., Kawamura, K., Carmichael, G. and Nakajima, T.: An overview of ACE-Asia: Strategies for quantifying the relationships between Asian aerosols and their climatic impacts, J. Geophys. Res.-Atmos., 108(D23), 8633, doi:10.1029/2003JD003550, 2003.

Liu, Z., Liu, D., Huang, J., Vaughan, M., Uno, I., Sugimoto, N., Kittaka, C., Trepte, C., Wang, Z., Hostetler, C. and Winker, D.: Airborne dust distributions over the Tibetan Plateau and surrounding areas derived from the first year of CALIPSO lidar observations, Atmos. Chem. Phys., 8(16), 5045–5060, 2008.

McKendry, I. G., Macdonald, A. M., Leaitch, W. R., van Donkelaar, A., Zhang, Q., Duck, T. and Martin, R. V.: Trans-Pacific dust events observed at Whistler, British Columbia during INTEX-B, Atmos. Chem. Phys., 8(20), 6297–6307, 2008.

Proestakis, E., Amiridis, V., Marinou, E., Georgoulias, A. K., Solomos, S., Kazadzis, S., Chimot, J., Che, H., Alexandri, G., Binietoglou, I., Daskalopoulou, V., Kourtidis, K. A., de Leeuw, G. and Ronald, J. van der A.: Nine-year spatial and temporal evolution of desert dust aerosols over South and East Asia as revealed by CALIOP, Atmos. Chem. Phys., 18(2), 1337–1362, doi:10.5194/acp-18-1337-2018, 2018.

Prospero, J. M., Ginoux, P., Torres, O., Nicholson, S. E. and Gill, T. E.: Environmental

characterization of global sources of atmospheric soil dust identified with the Nimbus 7 Total Ozone Mapping Spectrometer (TOMS) absorbing aerosol product, Rev. Geophys., 40(1), 1002, doi:10.1029/2000RG000095, 2002.

Shaw, G.: Transport of Asian Desert Aerosol to the Hawaiian-Islands, J. Appl. Meteorol., 19(11), 1254–1259, doi:10.1175/1520-0450(1980)019<1254:TOADAT>2.0.CO;2, 1980.

Sogacheva, L., Leeuw, G. de, Rodriguez, E., Kolmonen, P., Georgoulias, A. K., Alexandri, G., Kourtidis, K., Proestakis, E., Marinou, E., Amiridis, V., Xue, Y. and A, R. J. van der: Spatial and seasonal variations of aerosols over China from two decades of multi-satellite observations – Part 1: ATSR (1995–2011) and MODIS C6.1 (2000–2017), Atmospheric Chemistry and Physics, 18(15), 11389–11407, doi:https://doi.org/10.5194/acp-18-11389-2018, 2018.

Stith, J. L., Ramanathan, V., Cooper, W. A., Roberts, G. C., DeMott, P. J., Carmichael, G., Hatch, C. D., Adhikary, B., Twohy, C. H., Rogers, D. C., Baumgardner, D., Prenni, A. J., Campos, T., Gao, R., Anderson, J. and Feng, Y.: An overview of aircraft observations from the Pacific Dust Experiment campaign, J. Geophys. Res.-Atmos., 114, D05207, doi:10.1029/2008JD010924, 2009.

Tan, S.-C., Li, J., Che, H., Chen, B. and Wang, H.: Transport of East Asian dust storms to the marginal seas of China and the southern North Pacific in spring 2010, Atmos. Environ., 148, 316–328, doi:10.1016/j.atmosenv.2016.10.054, 2017.

Uno, I., Amano, H., Emori, S., Kinoshita, K., Matsui, I. and Sugimoto, N.: Trans-Pacific yellow sand transport observed in April 1998: A numerical simulation, J. Geophys. Res.-Atmos., 106(D16), 18331–18344, doi:10.1029/2000JD900748, 2001.

Uno, I., Yumimoto, K., Shimizu, A., Hara, Y., Sugimoto, N., Wang, Z., Liu, Z. and Winker, D. M.: 3D structure of Asian dust transport revealed by CALIPSO lidar and a 4DVAR dust model, Geophys. Res. Lett., 35(6), L06803, doi:10.1029/2007GL032329,

2008.

Xu, H., Zheng, F. and Zhang, W.: Variability in Dust Observed over China Using A-Train CALIOP Instrument, Adv. Meteorol., 1246590, doi:10.1155/2016/1246590, 2016.

Yumimoto, K., Eguchi, K., Uno, I., Takemura, T., Liu, Z., Shimizu, A. and Sugimoto, N.: An elevated large-scale dust veil from the Taklimakan Desert: Intercontinental transport and three-dimensional structure as captured by CALIPSO and regional and global models, Atmos. Chem. Phys., 9(21), 8545–8558, 2009.

Zhang, X. Y., Arimoto, R. and An, Z. S.: Dust emission from Chinese desert sources linked to variations in atmospheric circulation, J. Geophys. Res.-Atmos., 102(D23), 28041–28047, doi:10.1029/97JD02300, 1997.

Zhang, X. Y., Gong, S. L., Shen, Z. X., Mei, F. M., Xi, X. X., Liu, L. C., Zhou, Z. J., Wang, D., Wang, Y. Q. and Cheng, Y.: Characterization of soil dust aerosol in China and its transport and distribution during 2001 ACE-Asia: 1. Network observations, J. Geophys. Res.-Atmos., 108(D9), 4261, doi:10.1029/2002JD002632, 2003.

---

## Author Comment (AC1) · 13 Nov 2018

In this document, reviewer comments are in black, authors' response is in red, and the revised text is in blue.

On behalf of all authors, I would like to thank Referee #1 for their insightful suggestions and detailed corrections. We have made corresponding changes, which hopefully improve our manuscript. Responses to each point follow below.

Yan

General comments:

Yu et al. describe long-range transport of Asian dust with satellite observations and forward trajectories. The MINX tool was used to retrieve dust injection heights at the Gobi and Taklamakan deserts from measurements of the MISR instrument on board the Terra satellite. The obtained heights were consequently used to initiate HYSPLIT forward-trajectories, which were then used to describe the trans-Pacific dust transport and its seasonal and spatial variation. Such long-term data sets are useful for statistical analyses of dust layer heights and dust transport, needed for radiative transfer calculations and aerosol-cloud interaction studies. This is of high importance, especially for regions with sparse coverage of ground-based observations.

The manuscript is well written and I suggest publication in ACP after clarification of a few points (see specific comments) and some minor technical corrections.

Specific comments:

Concerning the temporal resolution of the MISR observations (crossing of the study region every 6-7 days), I have some questions about the dust events and plume data points: For example for the 2-3 years of analyzed data in the Gobi desert: Let us assume, 3 years, 1095 days. Now we have an observation every 6.5 days, so we have 168 observations. At 2251 dust events, this means 13 dust events per observation. How do you separate those dust events, source regions (intra-desert), and finally injection heights?

Thanks for carefully checking into our data. Dust plumes are identified through MINX from MISR radiance imagery by a trained user, with the assistance from the Support Vector Machine (SVM) datasets in the MISR cloud classifiers product. With MISR orbit information, MINX can identify the location, i.e. latitude and longitude, of the dust plume, and digitize the injection heights at each dust plume data point with 1.1 km spatial resolution. We added a brief description in the data section (P3 L27-32):

"Following Nelson et al. (2013), a "dust plume" is defined in this paper as a region of optically distinct dust that extends from an identified source to a downwind region, with a visible connection to the source, so that the direction of aerosol transport can be determined visually by the user. A "dust plume" typically contains hundreds to thousands of "dust plume data points". Dust plumes are identified through MINX from MISR radiance imagery by a trained user, with the assistance from the Support Vector Machine (SVM) datasets in the MISR cloud classifiers product (Nelson et al. 2013)".

Concerning injection heights: You state the method works independently of background aerosol and thin cirrus, but how do you deal with complex multilayer structures of (possibly) optically very dense dust layers?

Since MISR is an imaging spectroradiometer instrument, it is not capable of observing multilayer structures of optically thick dust layers. If the lower layers are not optically too thick and MISR can still see the land surface, MISR likely captures the height and motion of the top layer that has identifiable motion. We have clarified this point in the discussion section (P10 L30 – P11 L2):

"In addition, the current study focuses on trajectories initiated at the top of observed dust plumes. Given the capability of MISR at observing plume top features, we cannot infer the vertical structure of dust plumes from MISR stereo observations".

Concerning background aerosol: Do you consider or observe high-altitude layers of (polluted) dust

probably of western origin as described for example in Tanaka et al., 2005 (https://doi.org/10.1016/j.atmosenv.2005.03.034), Mikami et al., 2006 (https://doi.org/10.1016/j.gloplacha.2006.03.001), and Hofer et al., 2017 (https://doi.org/10.5194/acp-17-14559-2017)

According to the dust plume top height distributions in Fig. 3 and Fig. 4, our dataset apparently does not include such dust events several kilometers above the ground as discussed in these studies. Indeed, the greatest dust plume top height in the MINX dust plume dataset we have is 3021 m ASL over the Taklamakan Desert, and 2915 m ASL over the Gobi Desert. Therefore, we believe the dust plume observations we have analyzed mainly correspond to the dust emission from the Taklamakan and Gobi deserts. We briefly discuss this point in section 3.1 (P6 L7-9):

"The greatest dust plume top height in the MINX dataset is 3021 m ASL over the Taklamakan Desert, and 2915 m ASL over the Gobi Desert, apparently corresponding to local dust emissions, rather than from remote dust sources as reported in other studies (Hofer et al., 2017; Mikami et al., 2006; Tanaka et al., 2005)".

Concerning the difference in particle sizes of Gobi and Taklamakan dust: You mention this difference for soil (Sun et al., 2013), and that it probably has implications on the potential far range transport of Gobi dust compared to Taklamakan dust. However, what about the actual mobilization and air-borne dust if coarser particles were mobilized, as you state, at higher winds in Gobi? It could compensate to a certain degree the gravitational settling (for example, Gasteiger et al., 2017 (https://www.atmoschem-phys.net/17/297/2017/) try to explain the long-range transport (westward, though) of very coarse dust.

Thanks for the insightful comment. We discuss the soil particle size difference between the two deserts and the implication on long-range transport mainly through dust injection height, because our trajectory model does not address gravitational settling. But we totally agree that stronger surface wind that activates dust mobilization in the Gobi Desert likely triggers stronger vertical mixing, as revealed by Gasteiger et al. (2017) regarding dust transport in the Sahara Air Layer, thus compensating the gravitational settling to a certain degree. This motivates an interesting future study. We discuss this potential future study in the discussion section (P10 L24-26):

"However, the stronger surface wind that activates dust mobilization in the Gobi Desert likely triggers stronger vertical mixing, as revealed by Gasteiger et al. (2017) regarding dust transport in the Sahara Air Layer, thus compensating the gravitational settling to a certain degree.".

Technical corrections (minor formatting and typing errors):
Thanks for the technical corrections. We include these corrections in the revised manuscript.

Page 2 Line 18: In general, locked space between Fig. and number.
Page 2 Line 20: Tian Shan Mountain -> Tian Shan Mountains or Tian Shan Mountain system
Page 4 Line 27/26: leave "elevation" out or at least write ASL behind the values.
Page 6 Line 8: 15 m (line break) s^-1 -> use locked spaces ~ before (and within) units
Page 6 Line 34: p's -> p-values
Page 9 Line 2: "the wettest year in an land surface model ensemble simulation" -> "an" is wrong here, "any" or "a"?
Page 10 Line 34: Yu et al., 2017a, 2017b -> Yu et al. 2017a,b
Page 11 Line 21 : "Geosci. Remote Sensing, IEEE Trans." -> Abbreviation and comma are wrong, use "IEEE Trans. Geosci. Remote Sens."
Page 11 Line 29: add number (16) to volume 8. Furthermore, the doi is only here formatted as a web link, I think this is nowadays standard at ACP, use it everywhere.
Page 12 Line 15: Pages are wrong. The article has a page-like number (L06824), it might need to be stated, I don't know, but it has to be consistent within the References.

Page 12 Line 18: The article has a page-like number (L19802).
Page 12 Line 21/22: The article has a page-like number (D23212).
Page 12 Line 27: doi is wrong, digit missing and with a space. The correct one is 10.1002/2014JD021796. Check capitalization in the title, "East Asian".
Page 12 Line 32: The article has a page-like number (114018)
Page 12 Line 33: Rest of the authors is missing. Huebert, Bates, Russel, etc.
Page 13 Line 3: Here it is not "J. Geophys. Res. Atmos.", it is only "J. Geophys. Res.". Furthermore is the number D20 and the page-like number 9000.
Page 13 Line 4/6: "Kahn, R. a." -> "Kahn, R. A.". Probably this conference contribution is cited better like this (as it is stated on the SPIE homepage): "Proc. SPIE 8177, Remote Sensing of Clouds and the Atmosphere XVI, 81770O (26 October 2011)" plus doi and year, of course. Check the page numbers.
Page 13 Line 8: The article has a page-like number (L03106)
Page 12 Line 21: Why "(April 2006)"?
Page 12 Line 32/33: This reference is insufficient. Do you mean this?
https://eospso.gsfc.nasa.gov/sites/default/files/atbd/MISR_L3_CMV_ATBD.pdf Add the link and date of last access.
Page 14 Line 4: Journal is missing, it is: "Proc. SPIE 7089, Remote Sensing of Fire: Science and Application, 708909 (27 August 2008)"
Page 14 Line 5: "Kahn, R. a. and Dunst, B. a." -> "Kahn, R. A. and Dunst, B. A."
Page 14 Line 11: Take care of the special characters and capitalization. ACP provides a working bib file: AUTHOR = {Salvador, P. and Alonso-P\'erez, S. and Pey, J. and Art\'{\i}\~nano, B. and de Bustos, J.
J. and Alastuey, A. and Querol, X.},
Page 14 Line 16/17: Check capitalization. The title is: NOAA's HYSPLIT Atmospheric Transport and Dispersion Modeling System.
Page 14 Line 20: The article has a page-like number (D05207). Again „Atmos." is not necessary in the older JGR articles (I think before 2013), maybe even wrong, I am confused. Either you put „Atmos." everywhere (each issue D = Atmospheres) or nowhere in these articles.
Page 14 Line 26: "Atmos."? And in that specific article there are commas for digit grouping (strange, I know): 10,325-10,333. The same is in Huang et al., 2014, but there it is already correct. If you want to leave it out, you have to leave it out there as well.
Page 14 Line 29/30: Journal abbreviation is wrong, it should be "Geochem. Geophys. Geosyst."
Page 15 Line 4/5: "Atmos."? and the article has a page-like number (D17311)
Page 15 Line 9: "Atmos."? and the article has a page-like number (D12213).
Page 15 Line 11: "theglobe" -> "the globe".
Page 15 Line 23: "Tsay S. C." needs hyphen "Tsai S.-C." like in Wang et al., 2012b and the article has a page-like number (L08802).
Page 15 Line 27: Why (X)? Maybe (Part A).
Page 15 Line 29: Replace "n/a-n/a". The article has a page-like number (L05811).
Page 15 Line 33: "Atmos."? and the article has a page-like number (D00H35).
Page 16 Line 1: Why (April 1998)?
Page 16 Line 4: Here "Atmos." would fit.
Page 16 Line 6: (May) probably not needed, pages neither, article number is 15333. Like in Yu et al. 2017a.
Page 16 Line 21: No pages, the article has a page-like number (L07603).
Page 16 Line 23: No pages, the article has a page-like number (L18815).
Page 16 Line 25: No pages, the article has a page-like number (2272).
Page 17 Figure 1: "(%sample maximum)" -> "(% sample maximum)" or "(% of sample maximum)"
Page 17 Figure 2: Add ASL behind elevation values.
Page 18 Figure 4: The subfigures d) e) f) slightly overlap their titles. In general, the figures need a bit a

higher dpi for final publication.

Page 24 Figure 11: Caption, 10th, 25th etc. not superscript as in the caption of Fig. 2 and 4. Please be consistent.

Supplement Figure S3: Add ASL behind elevation values. And full stop or colon behind bold figure numbers in general.

In this document, reviewer comments are in black, authors' response is in red, and the revised text is in blue.

On behalf of all authors, I would like to thank Referee #2 for their valuable suggestions, especially those on results interpretation. We have made corresponding changes, which hopefully improve our manuscript. Responses to each point follow below.

Yan

The paper "Climatology in Asian dust activation and transport based on MISR satellite observations and trajectory analysis" presents and discusses the transport of dust aerosols, emitted from the arid and semiarid deserts of Taklamakan and Gobi, over the northern Pacific Ocean. The study falls within the scope of ACP. The study is based on MISR observations, MINX aerosol top height, and accordingly, forward HYSPLIT trajectory analysis. The manuscript is well-written/structured, the presentation clear, the language fluent. However, the submitted study is subject to major deficiencies and I would recommend publishing in ACP considering major revision.
Comments:
1) Regarding the "Asian dust activation climatology". Dust aerosol classification is crucial in the scope of the study, since it is the initial point of the trajectories analysis. Therefore, I would recommend to the authors to describe briefly the dust aerosol classification in MISR/MINX (including the necessary references). The scientific methods and assumptions are not clearly outlined. How is a "dust plume" defined in the paper and how is a "dust event"? In addition, in case of air parcels containing dust aerosols originating from both the Taklimakan and Gobi desert, how is the discrimination performed to the different sources? Which are the uncertainties in the classification?
We now describe briefly how dust plumes are identified in the current study (P3 L30-32):
"Dust plumes are identified through MINX from MISR radiance imagery by a trained user, with the assistance from the Support Vector Machine (SVM) datasets in the MISR cloud classifiers product (Nelson et al. 2013)".

We now define "dust plume" and "dust plume data point" clearly in the revised manuscript (P3 L27-30):
"Following Nelson et al. (2013), a "dust plume" is defined in this paper as a region of optically distinct dust that extends from an identified source to a downwind region, with visible connection to the source, so that the direction of aerosol transport can be determined visually by the user. A "dust plume" typically contains hundreds to thousands of "dust plume data points"".
"Dust event" was the same as "dust plume" in the previous version of the manuscript, and now are all changed to "dust plume".

In terms of the dust plume dataset, every dust plume retrieved by MINX has its identifiable source, either in the Taklamakan or Gobi Desert, as suggested by the definition of "dust plume". In terms of the trajectory analysis, we use forward trajectory so that the origin of any air parcel is either in the Taklamakan or Gobi Desert, depending on the trajectory starting point. There is possibility that two trajectories initiated from either desert merge. But the trajectory endpoint (Figs. 6 and 7) and trajectory passages (Figs. 8, 9, and 10) are analyzed for each desert separately, thereby such air parcels containing dust aerosols from both deserts will not affect the statistical analysis and conclusions presented in the current paper.

2) Regarding the "Asian dust transport climatology". Although the paper presents an interesting approach to study dust transport the results are not sufficient to support the conclusions, due to the lack of observations provided on parallel with the trajectories.

The study uses MISR observations-MINX provided top height to initiate HYSPLIT forward trajectories. Accordingly the climatology of trajectories is provided and not the Asian dust transport climatology. The difference is substantial. HYSPLIT computes the air parcel's transport and dispersion from a source region (Taklamakan and Gobi here) and describes where the air parcel will go. In the framework of the study, the climatology of the trajectories is provided (spatial distribution - % of trajectory endpoints / Trajectory passage frequency - % of trajectories after a specific number of days), without providing any observation/evidence on the presence of dust (per trajectory, distance or area). Dust aerosols may already have been removed along the transport/trajectory due to dry (gravitational settling) or wet deposition, although the air parcel will reach further distances. The paper does not even provide quantitative information on the probability of dust to have been transported. The trajectory may extend over the Pacific Ocean, and even further, to the western coast of the United States, however this does not provide any guarantee that dust is present and has reached that distance. I would suggest the authors to do any necessary modifications to the manuscript. Either provide dust observations per trajectory or to focus on the trajectories analysis without giving the impression on the presence (and transport) of dust to the trajectories endpoint. Which are the uncertainties? Alternatively, the authors could implement observations on the presence of dust to the western coast of USA (i.e. AERONET and AE, MODIS DT AOD and AE over ocean/ CALIOP volume/particle depolarization ratio) and use HYSPLIT back-trajectories. In addition, assuming a dust plume over an area, HYSPLIT initiated at different altitudes may provide different dust transport pathways. Therefore the study is representative only for the trajectories of the dust top-height and not for the dust plume (trajectories initiated at center of mass/scale height most probably would be more representative to discuss dust transport climatology).

Thanks for the valuable suggestion. We have been completely aware of the limitations in the trajectory analysis, and we have been very careful not to over-interpret results from the trajectory analysis. We also discussed the limitations of the current analysis in the discussion section of the original manuscript, including failure of considering wet or dry deposition and uncertainties caused by using the NCEP-NCAR reanalysis. We certainty agree that examining dust observations per trajectory will enable more robust conclusions regarding dust transport. We really appreciate reviewer's understanding of the amount of additional work regarding dust observations per trajectory. So we decide to go with the reviewer's suggestion about focusing on the trajectories analysis without giving too much impression on the presence (and transport) of dust to the trajectories endpoint. Therefore we revise the entire paper to focus on trajectory, such as replacing "dust transport" with either "dust trajectory" or "dust transport potential" throughout the paper. We also emphasize the uncertainty in the current study in the conclusion, such as (P10 L20-22) "Therefore, the current trajectory analysis provides an upper limit of the actual frequency of long-range dust transport, in particular the trans-Pacific dust transport from Asian sources to North America". We also expand the discussion on future work, adding sentences like (P10 L26-29) "These hypothesis can be tested by analyzing particle size distribution along trajectories from both deserts using ground and satellite observations, as well as performing advanced trajectory analysis that considers gravitational settling and wet deposition of dust particles", and (P11 L12-14) "In order to verify the identified seasonality in dust trajectory patterns, we suggest future studies to take advantage of both geostationary and polar-orbiting satellite observations, as well as ground-based lidar observations. Such dust observations per trajectory will eventually connect the trajectory analysis with actual dust transport.".

Regarding trajectory initial height, we now point out in the data section that the trajectory climatology presented in the current paper refers to that from dust plume top-height: "Given the capability of MISR at observing plume top features, we only analyse trajectories initiated at the observed dust plume top height in the current study". Given the capability of MISR, we are not able to infer the height of dust plume mass/scale center. In a future study, we will consider initiating trajectories using CALIOP or other lidar

observations of dust profiles. This is also briefly discussed in the discussion section of the revised manuscript (P10 L30 – P11 L2): "In addition, the current study focuses on trajectories initiated at the top of observed dust plumes. Given the capability of MISR at observing plume top features, we cannot infer the vertical structure of dust plumes from MISR stereo observations. With observed dust plume vertical structure, future studies are encouraged to analyze trajectories initiated at all vertical levels with the presence of dust aerosols".

3) The title does not reflect the contents of the paper and is misleading.
Corresponding to the limitations in trajectory modeling to infer dust transport, we change the title to "Climatology of Asian dust activation and transport potential based on MISR satellite observations and trajectory analysis".

4) The figures are of high quality. I would suggest on Figure 2 to reverse the axes, time on horizontal axis and Injection height at the vertical height.
We revised Fig. 2 according to the reviewer's suggestion. The revised figure is attached.

[Figure]

Figure 2: Atmospheric suspension time (hours) of dust particles emitted from the Taklamakan (40°N, 89°E, elevation = 805 m ASL) (blue) and Gobi Deserts (43.5°N, 130°E, elevation = 954 m ASL) (red) as a function of injection height (m ASL), based on trajectories in March-May of 2001-2003. The thick lines (shading) represent the median ($10^{th}$ to $90^{th}$ percentiles) of suspension time among 276 trajectories for each injection height.

5) Regarding Figure 3 and the Gobi Desert the lack of continuity in wind speeds between 3 and 12 m/s is a strange feature. I would suggest the authors to describe this feature.
Given the definition of "dust plume" in this study, the dust plume dataset contains dust plume data points both over the source and downwind. While dust particles are activated by strong surface wind (> 10 m s$^{-1}$ in Gobi, in this case), it is not necessary that strong wind persist to the downwind areas. This difference in wind speed over source and downwind regions is the most likely explanation of the discontinuity in wind speed distribution over the Gobi Desert. We now describe this feature in section 3.1 (P6 L17-20): "Indeed, since the dust plume dataset contains points both over and downwind of the source, dust particles from the Gobi Desert are usually activated by strong surface winds exceeding 10 m s$^{-1}$. The wind speed decreases quickly downwind of the actual source, causing the apparent discontinuity in the wind speed distribution over the Gobi Desert (Fig. 3b)".

6) Regarding references, a brief list of references is provided. I would suggest the authors to expand the list of references in order to strengthen the manuscript and at the same time to give credit to related work. Indicatively, here a brief list of related

studies is provided, describing features of dust aerosol transport emitted from the Taklamakan and Gobi deserts, based on synergies of passive and active ground-based and satellite-based instrumentation, models, campaigns and the meteorological and topographical mechanisms.

Thanks for the list of additional reference. We incorporate all the additional references in the revised manuscript.

Bory, A. J. M., Biscaye, P. E. and Grousset, F. E.: Two distinct seasonal Asian source regions for mineral dust deposited in Greenland (NorthGRIP), Geophys. Res. Lett., 30(4), 1167, doi:10.1029/2002GL016446, 2003.

Chen, S., Huang, J., Li, J., Jia, R., Jiang, N., Kang, L., Ma, X. and Xie, T.: Comparison of dust emissions, transport, and deposition between the Taklimakan Desert and Gobi Desert from 2007 to 2011, Sci. China-Earth Sci., 60(7), 1338–1355, doi:10.1007/s11430-016-9051-0, 2017.

de Leeuw, G., Sogacheva, L., Rodriguez, E., Kourtidis, K., Georgoulias, A. K., Alexandri, G., Amiridis, V., Proestakis, E., Marinou, E., Xue, Y. and van der A, R.: Two decades of satellite observations of AOD over mainland China using ATSR-2, AATSR and MODIS/Terra: data set evaluation and large-scale patterns, Atmos. Chem. Phys., 18(3), 1573–1592, doi:10.5194/acp-18-1573-2018, 2018.

Duce, R., Unni, C., Ray, B., Prospero, J. and Merrill, J.: Long-Range Atmospheric Transport of Soil Dust from Asia to the Tropical North Pacific - Temporal Variability, Science, 209(4464), 1522–1524, doi:10.1126/science.209.4464.1522, 1980.

Huebert, B. J., Bates, T., Russell, P. B., Shi, G. Y., Kim, Y. J., Kawamura, K., Carmichael, G. and Nakajima, T.: An overview of ACE-Asia: Strategies for quantifying the relationships between Asian aerosols and their climatic impacts, J. Geophys. Res.-Atmos., 108(D23), 8633, doi:10.1029/2003JD003550, 2003.

Liu, Z., Liu, D., Huang, J., Vaughan, M., Uno, I., Sugimoto, N., Kittaka, C., Trepte, C., Wang, Z., Hostetler, C. and Winker, D.: Airborne dust distributions over the Tibetan Plateau and surrounding areas derived from the first year of CALIPSO lidar observations, Atmos. Chem. Phys., 8(16), 5045–5060, 2008.

McKendry, I. G., Macdonald, A. M., Leaitch, W. R., van Donkelaar, A., Zhang, Q., Duck, T. and Martin, R. V.: Trans-Pacific dust events observed at Whistler, British Columbia during INTEX-B, Atmos. Chem. Phys., 8(20), 6297–6307, 2008.

Proestakis, E., Amiridis, V., Marinou, E., Georgoulias, A. K., Solomos, S., Kazadzis, S., Chimot, J., Che, H., Alexandri, G., Binietoglou, I., Daskalopoulou, V., Kourtidis, K. A., de Leeuw, G. and Ronald, J. van der A.: Nine-year spatial and temporal evolution of desert dust aerosols over South and East Asia as revealed by CALIOP, Atmos. Chem. Phys., 18(2), 1337–1362, doi:10.5194/acp-18-1337-2018, 2018.

Prospero, J. M., Ginoux, P., Torres, O., Nicholson, S. E. and Gill, T. E.: Environmental characterization of global sources of atmospheric soil dust identified with the Nimbus 7 Total Ozone Mapping Spectrometer (TOMS) absorbing aerosol product, Rev. Geophys., 40(1), 1002, doi:10.1029/2000RG000095, 2002.

Shaw, G.: Transport of Asian Desert Aerosol to the Hawaiian-Islands, J. Appl. Meteorol., 19(11), 1254–1259, doi:10.1175/1520-0450(1980)019<1254:TOADAT>2.0.CO;2, 1980.

Sogacheva, L., Leeuw, G. de, Rodriguez, E., Kolmonen, P., Georgoulias, A. K., Alexandri, G., Kourtidis, K., Proestakis, E., Marinou, E., Amiridis, V., Xue, Y. and A, R. J. van der: Spatial and seasonal variations of aerosols over China from two decades of multi-satellite observations – Part 1: ATSR (1995–2011) and MODIS C6.1 (2000–2017), Atmospheric Chemistry and Physics, 18(15), 11389–11407, doi:https://doi.org/10.5194/acp-18-11389-2018, 2018.

Stith, J. L., Ramanathan, V., Cooper, W. A., Roberts, G. C., DeMott, P. J., Carmichael, G., Hatch, C. D., Adhikary, B., Twohy, C. H., Rogers, D. C., Baumgardner, D., Prenni, A. J., Campos, T., Gao, R., Anderson, J. and Feng, Y.: An overview of aircraft observations from the Pacific Dust Experiment campaign, J. Geophys. Res.-Atmos., 114, D05207, doi:10.1029/2008JD010924, 2009.

Tan, S.-C., Li, J., Che, H., Chen, B. and Wang, H.: Transport of East Asian dust storms to the marginal seas of China and the southern North Pacific in spring 2010, Atmos. Environ., 148, 316–328, doi:10.1016/j.atmosenv.2016.10.054, 2017.

Uno, I., Amano, H., Emori, S., Kinoshita, K., Matsui, I. and Sugimoto, N.: Trans-Pacific yellow sand transport observed in April 1998: A numerical simulation, J. Geophys. Res.-Atmos., 106(D16), 18331–18344, doi:10.1029/2000JD900748, 2001.

Uno, I., Yumimoto, K., Shimizu, A., Hara, Y., Sugimoto, N., Wang, Z., Liu, Z. and Winker, D. M.: 3D structure of Asian dust transport revealed by CALIPSO lidar and a 4DVAR dust model, Geophys. Res. Lett., 35(6), L06803, doi:10.1029/2007GL032329, 2008.

Xu, H., Zheng, F. and Zhang, W.: Variability in Dust Observed over China Using A-Train CALIOP Instrument, Adv. Meteorol., 1246590, doi:10.1155/2016/1246590, 2016.

Yumimoto, K., Eguchi, K., Uno, I., Takemura, T., Liu, Z., Shimizu, A. and Sugimoto, N.: An elevated large-scale dust veil from the Taklimakan Desert: Intercontinental transport and three-dimensional structure as captured by CALIPSO and regional and global models, Atmos. Chem. Phys., 9(21), 8545–8558, 2009.

Zhang, X. Y., Arimoto, R. and An, Z. S.: Dust emission from Chinese desert sources linked to variations in atmospheric circulation, J. Geophys. Res.-Atmos., 102(D23), 28041–28047, doi:10.1029/97JD02300, 1997.

Zhang, X. Y., Gong, S. L., Shen, Z. X., Mei, F. M., Xi, X. X., Liu, L. C., Zhou, Z. J., Wang, D., Wang, Y. Q. and Cheng, Y.: Characterization of soil dust aerosol in China and its transport and distribution during 2001 ACE-Asia: 1. Network observations, J. Geophys. Res. Atmos., 108(D9), 4261, doi:10.1029/2002JD002632, 2003.